# Impact of implementation of front-of-package nutrition labeling on sugary beverage consumption and consequently on the prevalence of excess body weight and obesity and related direct costs in Brazil: An estimate through a modeling study

**Natália Cristina de Faria[1], Gabriel Machado de Paula Andrade[2], Cristina Mariano Ruas[3], Rafael Moreira Claro[4,5], Luíza Vargas Mascarenhas Braga[1], Eduardo Augusto Fernandes Nilson[5,6], Lucilene Rezende Anastácio[1]***

**1** Post-Graduate Program in Food Science, Faculty of Pharmacy, Universidade Federal de Minas Gerais (UFMG), Belo Horizonte, Brazil, **2** Chemical Engineering Department, Universidade Federal do Rio de Janeiro (UFRJ), Rio de Janeiro, Brazil, **3** Department of Social Pharmacy, Faculty of Pharmacy, Universidade Federal de Minas Gerais (UFMG), Belo Horizonte, Brazil, **4** Department of Nutrition, School of Nursing, Universidade Federal de Minas Gerais (UFMG), Belo Horizonte, Brazil, **5** Centre for Epidemiological Research in Nutrition and Health (NUPENS), University of Sao Paulo, Sao Paulo, Brazil, **6** Oswaldo Cruz Foundation (Fiocruz) Brasília, Brasilia, Brazil

* lucilene.rezende@gmail.com

## Abstract

### Rationale

Intake of sugary beverages has been associated with obesity and chronic non-communicable diseases, thereby increasing the direct health costs related to these diseases. Front-of-package nutrition labeling (FoPNL) aims to help consumers understand food composition, thereby improving food choices and preventing the development of such diseases.

### Objective

To estimate, over five years, the impact of implementing FoPNL in Brazil on the prevalence of excess body weight and obesity in adults who consume sugary beverages and the direct costs related to such problems.

### Methods

A simulation study to performed to estimate the effect of FoPNL implementation on the prevalence of excess body weight and obesity. The VIGITEL research database (2019), published in the 2020 report, was used in this study (the final sample consisted of 12,471 data points representing 14,380,032 Brazilians). The scenarios were considered: base (trend in sugary beverage intake); 1 (base scenario associated with the changes in energy content of the purchased beverages observed after the first phase of the Chilean labeling law (−9.9%);

**Data Availability Statement:** https://svs.aids.gov.br/download/Vigitel/.

**Funding:** Conselho Nacional de Desenvolvimento Científico e Tecnológico-CNPq and Ministério da Saúde-MS (442990/2019-7) and Fundação de Amparo à Pesquisa do Estado de Minas Gerais-FAPEMIG (APQ-00341-21). Pro-Reitoria de Pesquisa da Universidade Federal de Minas Gerais. Coordenação de Aperfeiçoamento de Pessoal de Nível Superior (Capes). The funders had no role in study design, data collection and analysis, decision to publish, or preparation of the manuscript.

**Competing interests:** The authors have declared that no competing interests exist.

and 2 (scenario 1 associated with reformulation of beverages, total energy reduction of −1.6%). Changes in body weight were estimated using the simulation model of Hall et al. (2011) over five years. A linear trend in the prevalence of obesity and excess body weight in the Brazilian population was considered. The impact of the prevalence of obesity and excess body weight on body mass index was estimated. In addition, the direct health costs related to obesity were estimated.

## Results

Energy consumption from sugary beverages after FoPNL implementation is expected to be reduced by approximately 28 kcal/day (95% CI, −30 to −27) considering scenario 1. In scenarios 1 and 2, without FoPNL, the prevalence of obesity and excess body weight over five years was estimated to be 25.3% and 25.2%, and 64.4% and 64.2%, respectively. By extrapolating the results to the entire Brazilian population, it was observed that the implementation of FoPNL may reduce the prevalence of obesity by −0.32 percentage points and −0.35 percentage points (scenario 1 and 2, respectively) and excess body weight by −0.42 percentage points and −0.48 percentage points (scenarios 1 and 2, respectively) in five years. It is estimated that after five years of implementation, it will be possible to save approximately US$ 5,5 millions (95% CI 4,7 to 8,8) in scenario 1, reaching approximately US$ 6,1 millions (95% CI 5,3 to 9,8) in scenario 2.

## Conclusion

The results of this modeling study indicate that FoPNL may reduce prevalence of excess body weight and obesity, representing strategic public policies for obesity prevention.

## Introduction

Sugary beverages, such as soft drinks and fruit-based drinks, are well recognized for their deleterious effects on health [1]. These drinks contain high levels of free sugars [2] and represent the largest source of sugar consumption [3, 4]. In addition, these beverages, which are based on a mixture of water, some types of sugar, and a flavoring syrup, have low nutritional density and compromised diet quality [5]. Thus, such drinks are strongly associated with excessive weight gain, obesity [6], and chronic diseases such as type 2 diabetes melitus [1].

Although the consumption of sugary beverages has decreased in Brazil in recent years, a significant portion of the population still consumes these beverages every day [7]. A study in 2016 reported that one in six adults in state capitals and the Federal District consumes these beverages every day, which justifies interventions to maintain this reduction at even lower levels [7].

Obesity and its effects on health generated an estimate annual costs of R$ 1,42 billion (95% CI, 0,98–1,87) in 2018 *via* the Public Health System (SUS) in Brazil [8]. The costs included hospitalizations, outpatient procedures, and medications distributed by the SUS for the treatment of these diseases, excluding supplementary health costs in the country, as well as the economic and social costs associated with illness and death from these causes [8]. Direct costs are those paid by health services related to immediate expenses, and include labor, tests, and medications [9].

The implementation of front-of-package nutrition labeling (FoPNL) has been proposed as a public policy to reduce the consumption of unhealthy foods [10, 11]. The consumers can

easily identify foods and beverages that are dense in calories and added sugars and have reduced nutritional value. Consequently, by decreasing the consumption of unhealthy foods and beverages, the incidences of diet-related chronic non-communicable diseases can be reduced [12]. A meta-analysis identified that FoPNL was helpful in reducing energy consumption (corrected standardized difference = −0.16; 95% CI, −0.24 to −0.07; n = 2,338; p = 0.010) and sugar (g) procurement (corrected standardized difference = −0.11; 95% CI, −0.21 to 0.01; n = 1,938; p<0.001) when compared to the control (individuals Canadians and Americans were included in analysis) [13].

In Chile, the food industry reformulated foods and beverages after implementing FoPNL and other public health policies, reducing added sugars in sugary beverages and introducing non-nutritive sweeteners [14, 15]. Kanter et al. found a reduction in the content of added sugars in beverages between 2015 and 2016 (median reduction 7.5 g/100 mL, interquartile range (IQR): 2.3–10.0 in 2015 to 6.0 g/100 mL, IQR: 2.2–10 in 2016) in Chile, referring to reformulation [14].

Some countries such as Chile [16], Mexico [17], Canada [18], Brazil [19, 20] and other countries [21] have already implemented or in the process of implementing such policies. In 2016, Chile implemented Laws No. 20.606 and 20.869 on Food Labeling and Advertising, respectively [12]. In addition, FoPNL products cannot be marketed in schools for children under 14 years of age and promoted to this audience [22]. In addition to these policies, Chile has implemented tax legislation on sugary beverages containing greater than 15 g of sugar per 240 mL of beverage or equivalent portions containing more than 18% of sugars and other beverages containing 10% of sugar, intending to reduce their consumption [23]. Thus, several factors can influence consumer behavior [24].

After the implementation of Chilean legislation, which began in 2016, a decrease of 23.7% (95% CI, −23.8 to −23.7%) was reported in the purchase of sugary beverages with "high in" FoPNL [25]. A modeling study conducted in Mexico estimated that FoPNL can promote a reduction of 23.2 kcal/day (95% CI, −24.5 to −21.9) associated with the consumption of sugary beverages and 13.6 kcal/day (95% CI, −14.1 to −13.1) to snacks, resulting in 4.98 percentage points reduction in the number of obese people in the country [26]. Given this context, implementing FoPNL may be a useful strategy to reduce the consumption of sugary beverages, among other food products, as well as one of the strategies to prevent obesity. In October 2020, Brazil passed legislation that determined the use of FoPNL in the magnifying glass model on foods with a high content of critical nutrients (added sugars, saturated fats, and sodium) [19, 20]. However, although the impact of Brazilian FoPNL on food purchase intentions has been investigated in the Brazilian population [27–29], further studies are needed. In addition, investigations are needed to understand consumer behavior because of the implementation of FoPNL, such as possible changes in purchasing patterns, consequent changes in food consumption and causing changes in nutritional status. To the best of our knowledge, no study has simulated the effect of FoPNL on the prevalence of excess body weight and obesity in Brazil.

Thus, the objective of this study was to estimate, over five years, the reduction in the prevalence of obesity and excess body weight among Brazilian adults and the direct costs in public health system related to such problems after the implementation of FoPNL in Brazil.

## Method

### Study design

This economic impact assessment study was carried out from the perspective of the Brazilian public health system. A simulation model was used to estimate future impacts on the

prevalence of obesity and excess body weight, and the direct costs of obesity, which could be reduced by implementing FoPNL.

Three scenarios were considered over five years: i. base scenario: assessment of the prevalence of obesity and excess body weight and the direct costs of obesity in the absence of any health policy, using a time trend of obesity prevalence; ii. scenario 1: estimation of the prevalence of obesity and excess body weight and the direct costs of obesity after implementation of FoPNL, based on the change in the purchase of beverages observed after the first phase of the Chilean labeling law [30]. iii. scenario 2: estimation of the prevalence of obesity and excess body weight and the direct costs of obesity in scenario 1 associated with beverage reformulation in Chile as observed by Kanter et al. [14].

For scenarios 1 and 2, the possible reduction in calories from the consumption of sugary beverages after implementation of FoPNL was estimated. Further, the impact of the reduction in caloric intake on body weight and body mass index (BMI), and subsequently on the prevalence of obesity and excess body weight was estimated over five years.

## Study population, sampling, and ethical aspects

The VIGITEL (Surveillance System for Risk and Protective Factors for Chronic Diseases by Telephone Survey) database 2019 and published in the 2020 report was used to conduct the study [31]. This annual survey was conducted by the Ministry of Health for investigating the risk and protective factors for chronic non-communicable diseases since 2006. The VIGITEL was used in 26 capitals of Brazilian states and the Federal District from a probabilistic sample of the adult population residing in households with a landline telephone system. Additional information regarding the VIGITEL sampling and data collection process is available in a specific publication referring to the 2019 Report [31]. The VIGITEL 2019 questionnaire assessed demographic and socioeconomic characteristics, dietary pattern associated with the development of non-communicable chronic diseases (such as consumption of sugary beverages), and self-reported weight and height, among others. A total of 52,443 individuals were interviewed in the 2019 edition of the survey [31].

A data subsample from the VIGITEL 2019 comprising individuals of both sexes aged between 20 and 59 years was used for the present study. Individuals who reported never or almost never consumption of sugary beverages, who did not know how to report the amount consumed, who consumed only diet/light/zero beverages, pregnant women, and individuals with extreme BMI (greater than 60 kg/m$^2$ and less than 15 kg/m$^2$) were excluded (S1 Table). The final sample consisted of 12,471 data points representing 14,380,032 Brazilians (the projected value indicated the sum of the weighting factors of this population). The VIGITEL was approved by the National Commission for Ethics in Research with Human Beings of the Ministry of Health (CONEP Opinion 355590 of June 26, 2013, under CAAE number 16202813.2.0000.0008). Free and oral informed consent was obtained at the first instance of telephone contact with the participant. The survey data were publicly accessible and used without identification of the interviewed individuals [31]. The databases for all years of Vigitel's achievements are available at http://svs.aids.gov.br/download/Vigitel/. More details regarding VIGITEL are provided in the S1 File.

## Assessment of the intake of sugary beverages by the Brazilian population

Assessment of the consumption of sugary beverages was based on three questions from the VIGITEL survey: 'How many days a week do you usually drink soft drinks or artificial juices?' (1–2 days a week; 3–4 days a week; 5–6 days a week; every day, including Saturdays and Sundays; almost never; never), Which type? (normal; diet/light/zero; both), and 'How many

glasses/cans do you usually drink every day?' (1, 2, 3, 4, 5, 6, or more; I don't know). The questions related to soft drink consumption used in the VIGITEL survey passed through a previous validation analysis with reasonable values of specificity (all: 94.1%) and sensitivity (all: 87.5%) [32]. According to the Technical Regulation in Brazil, soft drinks are carbonated beverages obtained by dissolving the juice or plant extract in drinking water, added sugar, and saturated with industrially pure carbon dioxide. In this study, only soft drinks were considered as sugary beverages [33]. More details are provided in the S1 File.

## Estimation of reduction in energy intake

Several scenarios were used to estimate the reduction in energy intake in participants who reported drinking sugary beverages after implementation of the FoPNL (S3 Fig, S1 File). For the "no change" scenario, called the base scenario, the time trend of consumption observed by the VIGITEL between 2007 and 2019, except 2017, was calculated using linear regression (S4 and S5 Figs, S1 File). The databases of previous studies were adjusted using the same inclusion and exclusion criteria as described in the subsection Study population, sampling, and ethical aspects (n = 23,170 in 2007, n = 24,604 in 2008, n = 24,467 in 2009, n = 24,856 in 2010, n = 24,311 in 2011, n = 22,353 in 2012, n = 22,835 in 2013, n = 16,475 in 2014, n = 16,387 in 2015, n = 17,033 in 2016, n = 12,999 in 2018, and n = 12,471 in 2019). Further details are available in the S1 File. The time trend of consumption was used in the base scenario, whereas in the other scenarios, it was associated with reduction in energy intake through labeling and reformulation.

In scenario 1, we considered the results of the changes in calories and sodium content of beverage purchased after the first phase of implementation of the Chilean law, which includes the implementation of FoPNL [30]. Taillie et al. evaluated the food and beverage purchased by 2,381 Chilean families between 2015 and 2017, and compared purchases in the pre- and post-implementation periods of FoPNL. There was a reduction in the purchase of drinks labeled "high in", with some compensation in the purchase of drinks without FoPNL, with a total decline equivalent to 9.9% in calories and 5.2% in sodium (S4 and S5 Tables, S1 File) [30]. These results were associated with the current trend of reduction in the consumption of sugary drinks in Brazil, calculated in the base scenario, to compose scenario 1.

The reformulation of beverages immediately after the FoPNL implementation was also considered for the construction of scenario 2. In Chile, between 2015 and 2016, Kanter et al. [14] (S6 Table, S1 File) found a reduction of 1.6% in the sugar content contributing to the total energy to beverages (median: 30 kcal/100 mL, IQR: 12–44 in 2015 to 28 kcal/100 mL, IQR: 11–44 in 2016) [14]. However, sodium levels increased in beverages (median 10 mg/100mL, IQR 6–17 in 2015 to 10 mg/100mL, IQR 5–17 in 2016), with a mean difference of 1.8% [14]. In scenario 2 of this study, the trend in the consumption of sugary beverages in Brazil was considered, as described in the base scenario, in addition to the effect of the implementation of FoPNL observed by Acton et al. [30], associated with the effect of product reformulation demonstrated by Kanter et al. [14]. More details are provided in the S1 File.

The effect of reduction in the energy intake from sugary beverages related to the changes in FoPNL disregarded saturated fats and sugars to estimate body weight variation.

## Estimation of reduction in body weight, BMI, and prevalence of obesity and excess body weight

The impact on body weight, BMI, and the prevalence of obesity and excess body weight was estimated from the results obtained for the reduction in energy intake using the model proposed by Hall et al. [34]. This model used data on energy and sodium intake variation and

estimated body weight variation in each individual over time [34]. Therefore, body weight variation over five years was estimated based on the estimated energy intake change promoted by FoPNL. The model by Hall et al. considered variations in extracellular fluid, glycogen, adipose, and lean tissues, keeping physical activity constant [34]. The model considered age, time, initial body weight, height, and variation in energy/sodium intake. The body weight variation due to changes caused by the intake of sugary beverages was estimated [34], and BMI with new body weight was calculated.

Based on the BMI, the subjects were then classified as obese/non-obese or with/without excess body weight. We used the cutoff point adopted by the World Organization Health (WHO) for the classification of obese individuals: BMI $\geq$ 30 kg/m$^2$ [35]. Individuals with BMI $\geq$ 25 kg/m$^2$ were considered as having excess body weight. The percentage change in BMI was calculated as the difference between the number of obese individuals at the initial time and the estimated final time (S1 File).

## Reduction in the prevalence of obesity and excess body weight

The projection of the Brazilian population of adults aged between 20 and 59 years was verified over a period of five years (2020–2024), based on information contained in the Brazilian Institute of Geography and Statistics (IBGE) [36]. The proportion of Brazilians who consume sugary beverages was calculated based on information from previous years from the VIGITEL (2007 to 2019, except 2017) and projected to the Brazilian population from 2020 to 2024 (aged 20 to 59 years). According to the data obtained from the VIGITEL, in the same subsample from 2007 to 2019, except 2017, the prevalence of obesity and excess body weight among consumers of sugary beverages was calculated using linear regression analysis (S7 and S8 Figs, S8 Table, S1 File).

The change in the prevalence of obesity and excess body weight obtained in the previous step was multiplied by the estimated number of Brazilian consumers of sugary beverages. The difference in the number of individuals with obesity/excess body weight between scenarios 1 and 2 and the base scenario was calculated. The result obtained showed a decreasing trend in the obesity prevalence, and thus, a new prevalence of obesity and excess body weight was estimated for different study scenarios. The same was done with the estimate of population with excess body weight. The change in the prevalence of obesity for the total population, stratified by gender and excess body weight, was calculated. More details are provided in the S1 File.

## Data analysis and sensitivity analysis

The analyzed population was described according to characteristics related to sex and age and presented as the mean and 95% confidence interval (95% CI). The prevalence of sugary beverage consumers and their average daily consumption, stratified by sex, were estimated for each year using the VIGITEL survey. The consumption of sugary beverages was presented as the average and 95% CI. Linear trends were investigated using linear regression analysis. The dependent variables evaluated for each year included the annual proportion of Brazilians who consumed sugary beverages, average amount of sugary beverages consumed (in terms of energy and sodium), annual proportion of obese Brazilians who consumed sugary beverages, and annual proportion of Brazilians with excess body weight who consumed sugary beverages. The independent variables included data of the VIGITEL survey or all years used in this study. Regression models were stratified by sex, considering the sample weights of the VIGITEL survey. Thus, temporal trends (2007–2019, except 2017) were analyzed, and the projected prevalence (2020–2024) was calculated in the base scenario. Additional details are provided in the S1 File.

The range of uncertainty regarding the impact of FoPNL implementation on the prevalence of obesity and excess body weight was determined using sensitivity analysis. For this purpose, alternative scenarios were simulated. The scenario 3 impact promoted by FoPNL in adults (n = 1,213) regarding the purchase of sweetened beverages observed in the Canadian experimental market study (energy: −10.5%; sodium: −5.5%, S7 Table) was used [37, 38], which was consistent with the trend of consumption of sugary beverages in the Brazilian population. Acton et al. obtained their results through an experimental market study conducted in Canada using a "high in" FoPNL in a red circle (S6 Fig, S1 File) [37]. Finally, scenario 4 was based on reformulation of beverages (energy: −1.6%; sodium: 1.8%) observed by Kanter et al. in Chile [14], and was considered to be associated with scenario 3 described above.

## Cost estimation

The costs related to obesity were obtained based on a study by Nilson et al. [8]. This study describes the value of direct costs of SUS attributed to obesity in 2018 (S11 Table, S1 File). For this study, data referring to the direct costs of obesity in Brazilians aged 20–59 years were obtained from the study by Nilson et al. (2020).

With the total costs available, the per capita cost of obesity was obtained by using information on the estimated Brazilian population according to the data provided by the Brazilian Institute of Geography and Statistics for the year 2018 [36], concerning the year of the estimates of the costs described above [8].

The number of Brazilians who depend on the public health system [39] was estimated for 2018, the same year when the study database of Nilson et al. [8] was used to estimate the direct costs. Thus, the total number of obese people assisted by the public health system was estimated. It is estimated that 71.5% of the population depend on the public health system in Brazil [39]. The number of obese individuals assisted by the public health system was estimated by projecting the total population, stratified by sex. Thus, direct costs per capita were calculated using the ratio between the direct costs for 2018 and the number of obese people in the same age group (20–59 years) who were served by the public health system. The value per capita for 2019 was updated using the Broad National Consumer Price Index [40]. With the per capita cost corrected until 2019, the value was converted to US dollars (conversion rate: 1 USD = R$ 4.03). In addition, purchasing power parity was used as a conversion factor to compare the results (conversion rate of 2.281 in 2019). After this correction, a discount rate of 5% per year was applied to future values (2020–2024), a value recommended by the Brazilian Economic Valuation Guidelines [41]. Additional details are provided in the S1 File. Thus, to estimate the impact on direct health costs, the estimated per capita costs of obesity were multiplied by the number of cases of obesity that could be reduced after the implementation of FopNL in Brazil.

## Results

### Impact on the consumption of sugary beverages

The age of the individuals participated in the study was between 20 and 59 years (mean age 36.1 years; 95% CI, 35.8–36.5), 54.3% (95% CI, 52.7–56.0) of which were male (mean age 35.7 years; 95% CI, 35.2–36.2) and 45.7% (95% CI, 44.0–47.3%) were female (mean age 36.7 years; 95% CI, 36.2–37.2). It was observed that the average consumption of sugary beverages was 251.5 mL/day (95% CI, 240.6–262.5) in 2019. Higher average consumption of sugary beverages was observed in men than in women, 283.3 mL/day (95% CI, 265.6–303.1) and 213.7 mL/day (95% CI, 199.8–222.5), respectively.

**Table 1. Estimated change in the intake of sugary beverages over five years, in calories per day per person, after FoPNL implementation among Brazilian adults aged 20 to 59 years.**

| Intake of sugary beverages at the beginning of the study and estimated times | All | Male | Female |
|---|---|---|---|
| | **Average kcal per day per person (95% CI)** | | |
| **Base scenario** | −20 (−21 to −19) | −24 (−26 to −23) | −16 (−17 to −15) |
| Consumption trend of sugary beverages for the next five years | | | |
| **Scenario 1** | −28 (−30 to −27) | −33 (−35 to −31) | −23 (−24 to −21) |
| Expected change in the energy intake after five years of FoPNL implementation | | | |
| **Scenario 2** | −29 (−31 to −28) | −34 (−36 to −32) | −24 (−26 to −22) |
| Expected change in the energy intake after FoPNL implementation and reformulation | | | |

FoPNL: front-of-package nutrition labeling

After the implementation of FoPNL, it is estimated that energy consumption from sugary beverages was reduced by approximately −28 kcal/day (95% CI, −30 to −27) in scenario 1 and by 29 kcal (95% CI, −31 to −28) in scenario 2. According to the reduction trend in the consumption of sugary beverages observed in Brazil, it is estimated that the reduction will be approximately −20 kcal/day (95% CI, − 21 to −19) in 2024. Information on the estimated change energy consumption of sugary beverages is presented in Table 1.

## Impact on body weight reduction and prevalence of obesity and excess body weight

The body weight and mean BMI of the participating individuals were 75.4 kg (95% CI, 74.7–76.0) and 26.5 kg/m$^2$ (95% CI, 26.3–26.7), respectively. A reduction in the body weight of Brazilians by an average of 1.1 kg (95% CI, 1.0–1.1) has been estimated after five years of implementing FoPNL in scenario 1 and 1.1 kg (95% CI, 1.1–1.2) in scenario 2. This reduction would lead to an average BMI of 26.1 kg/m$^2$ (95% CI, 25.9–26.3) in Brazilian adult population five years after implementing FoPNL in scenario 1 (Table 2). By extrapolating the results for the entire Brazilian population, it was observed that implementation of FoPNL may reduce the prevalence of obesity and excess body weight by −0.32 percentage points (95% CI, −0.32 to −0.31) (men: −0.25 percentage points, 95% CI, −0.25 to −0.24; women: −0.37percentage points, 95% CI, −0.38 to −0.37) (Fig 1B-1) and −0.42 percentage points (95% CI, −0.42 to −0.41) (men: 0.55 percentage points, 95% CI, −0.56 to −0.55; women: −0.30 percentage points, 95% CI, −0.31 to 0.30) (Fig 1D-1). Similar values for the reduction in the prevalence of obesity and excess body weight were estimated for scenario 2. This reduction would be equivalent to approximately 391,529 cases of obesity after five years of implementing FoPNL in scenario 1 and 433,292 in scenario 2 (S9 and S10 Tables, S1 File). However, the reduction of simulated cases is unable to incline the curve of the increase in the prevalence of obesity, but it has the potential to slow down the growth.

## Impact on direct costs

Implementation of FoPNL for five years has been estimated to reduce direct costs of public health service in Brazil by approximately US$ 5,5 millions (95% CI 4,7 to 8,8) related to chronic non-communicable diseases associated with obesity in scenario 1, according to the model proposed as described in Fig 2. In scenario 2, an estimation of approximately US$ 6,1 (95% CI 5,3 to 9,8) millions has been projected.

**Table 2. Initial parameters and estimated change in body weight and body mass index after five years of implementation of FoPNL among Brazilian adults aged 20 to 59 years.**

| Evaluated parameters | All | Male | Female |
|---|---|---|---|
| Average (95% CI) | | | |
| **Base scenario**<br>Body mass index (kg/m$^2$) in 2019 | 26.5 (26.3–26.7) | 26.5 (26.3–26.7) | 26.5 (26.3–26.8) |
| **Base scenario**<br>Prevalence of obesity in 2019 (%) | 21.5 (20.3–22.7) | 20.1 (18.2–22.1) | 23.1 (21.3–25.0) |
| **Scenario 1**<br>Estimated change in body weight (kg) after five years of FoPNL implementation | −1.1 (−1.0 to −1.1) | −1.2 (−1.2 to 1.3) | −0.9 (−0.9 to −1.0) |
| **Scenario 1**<br>Estimated change in body mass index (kg/m$^2$) after five years of FoPNL implementation | 26.1 (25.9–26.3) | 26.1 (25.8–26.3) | 26.2 (25.9–26.5) |
| **Scenario 2**<br>Estimated change in body weight (kg) after five years of FoPNL implementation | −1.1 (−1.1 to −1.2) | −1.3 (−1.2 to −1.4) | −1.0 (−0.9 to −1.0) |
| **Scenario 2**<br>Estimated change in body mass index (kg/m$^2$) after five years of FoPNL implementation | 26.1 (25.9–26.3) | 26.0 (25.8–26.3) | 26.2 (25.9–26.5) |

FoPNL: front-of-package nutrition labeling

## Discussion

The study investigated the impact of FoPNL on Brazilian consumers of sugary beverages. In scenario 1, it is estimated that FoPNL implementation can reduce the energy intake by approximately −28 kcal/day (95% CI, −30 to −27) and body weight by −1.1 kg (95% CI, −1.0 to −1.1) over five years. This reduction in energy intake has the potential to decrease the prevalence of obesity and excess body weight (−0.32 percentage points and −0.41 percentage points, respectively) in Brazilian adults aged between 20 and 59 years after five years of FoPNL implementation. Such a reduction in the prevalence of obesity and excess body weight could reduce direct healthcare costs related to high BMI by approximately US$ 5,5 (95% CI 4,7 to 8,8) millions after five years. Since the consumption of sugary beverages is associated with the development of obesity and chronic non-communicable diseases, political strategies to reduce their consumption are important [6, 42].

Given the outcomes of food and beverage reformulation in Chile, a scenario considering reformulation was also chosen in this study. However, conservative reformulation scenario observed by Kanter et al. was selected because the cutoff points of the critical nutrient profile between Chile and Brazil are different (total sugar in beverages ≥ 6 g per 100 mL in the first phase of implementation; added sugar in beverages ≥ 7.5 g per 100 mL) [19, 20, 43]. Moreover, measures such as beverage taxation, rules related to advertising, and sales policies for food with FoPNL have been adopted in Chile, which has not yet occurred in Brazilian legislation [23]. Although there are differences in nutritional profile and public health policies between the two countries, reformulation of soft drinks has probably occurred in Brazil since non-sugar sweeteners seem to have been frequently used in these beverages [44].

In scenario 2, the possibility of reformulation of sugary beverages was estimated; thus, the reduction in energy intake from sugary beverages was expected to be slightly higher than 29 kcal/day (95% CI, −31 to −28). However, there is concern about this practice given the possible negative health effects associated with the intake of non-sugar sweeteners [45].

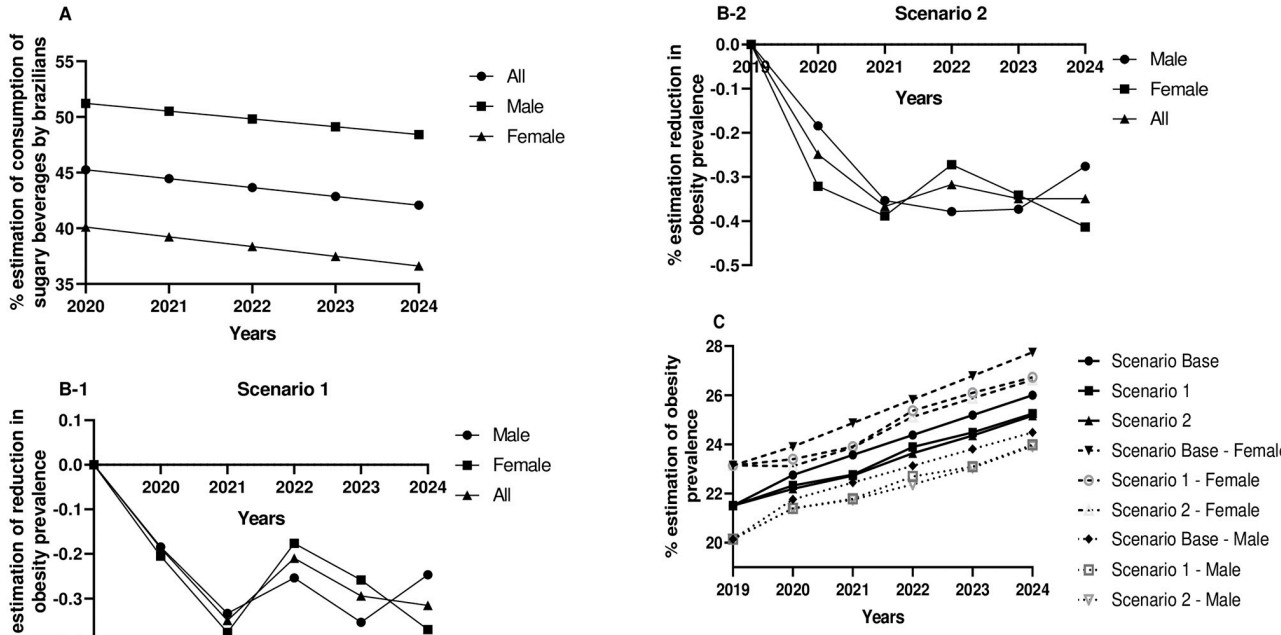

**Fig 1. Estimation of the percentage and change in the prevalence of obesity and excess body weight over five years among Brazilian adults aged 20 to 59 years and sugary beverage consumers, stratified by gender, in different study scenarios.** A: Estimation of the proportion of sugary beverages consumed by Brazilians. B: Estimation of the proportion of reduction in obesity prevalence in scenario (1) and scenario 2 (2) in the Brazilian population. C: Estimation of obesity prevalence in all consumers of sugary beverages (both males and females) in five years in the base scenario, scenarios 1, and scenario 2. D: Estimation of the proportion of reduction in excess body weight prevalence in scenario 1 (1) and scenario 2 (2) in the Brazilian population. E: Estimation of excess body weight prevalence in all consumers of sugary beverages (both male and female) in five years in the base scenario, scenarios 1, and scenario 2.

In the present study, it was estimated that implementation of FoPNL could reduce the prevalence of obesity by 0.32 percentage points. Furthermore, the scenarios were developed considering the trend of reduced intake of sugary beverages in Brazil [7] as well as the increasing prevalence of obesity/excess body weight. The FoPNL model used in Chile is the FoPNL "high in sugar" warning label in black octagon (S2 Fig, S1 File) for beverages containing more than 6 g of sugar per 100 mL (S3 Table, S1 File). In Brazil, beverages with added sugars receive a FoPNL in the form of a magnifying glass (S1 Fig, S1 File) if they contain more than 7.5 g of added sugar per 100 mL (S2 Table, S1 File) [20].

The design of the FoPNL adopted in Brazil (black magnifying glass), unlike an octagon, circle, or triangle, is not considered a warning [46]. There is little evidence for the efficiency of the model adopted in Brazil, although it seems to help identify excess nutrients [29]. Brazilian and Mexican labeling systems were evaluated in Brazilian population aged more than 18 years in a supermarket using a smartphone application. It was observed that the Mexican system performed better than the Brazilian system in identifying excess added sugars and in purchase intentions [27]. Prates et al. (2022) also showed that the perception of healthiness and purchase intention was higher with FoPNL in a magnifying glass format, compared to the warning model, in Brazilians aged over 18 years in an online survey [28]. Moreover, it was observed that the nutrient profile adopted in Brazil labels a smaller number of products, compared to the Mexican FoPNL system [44].

Implementing FoPNL can be an efficient public health strategy because it may reduce mortality of chronic non-communicable diseases [47]. The decrease of 0.32 percentage points observed in the prevalence of obesity and 0.42 percentage points in excess body weight

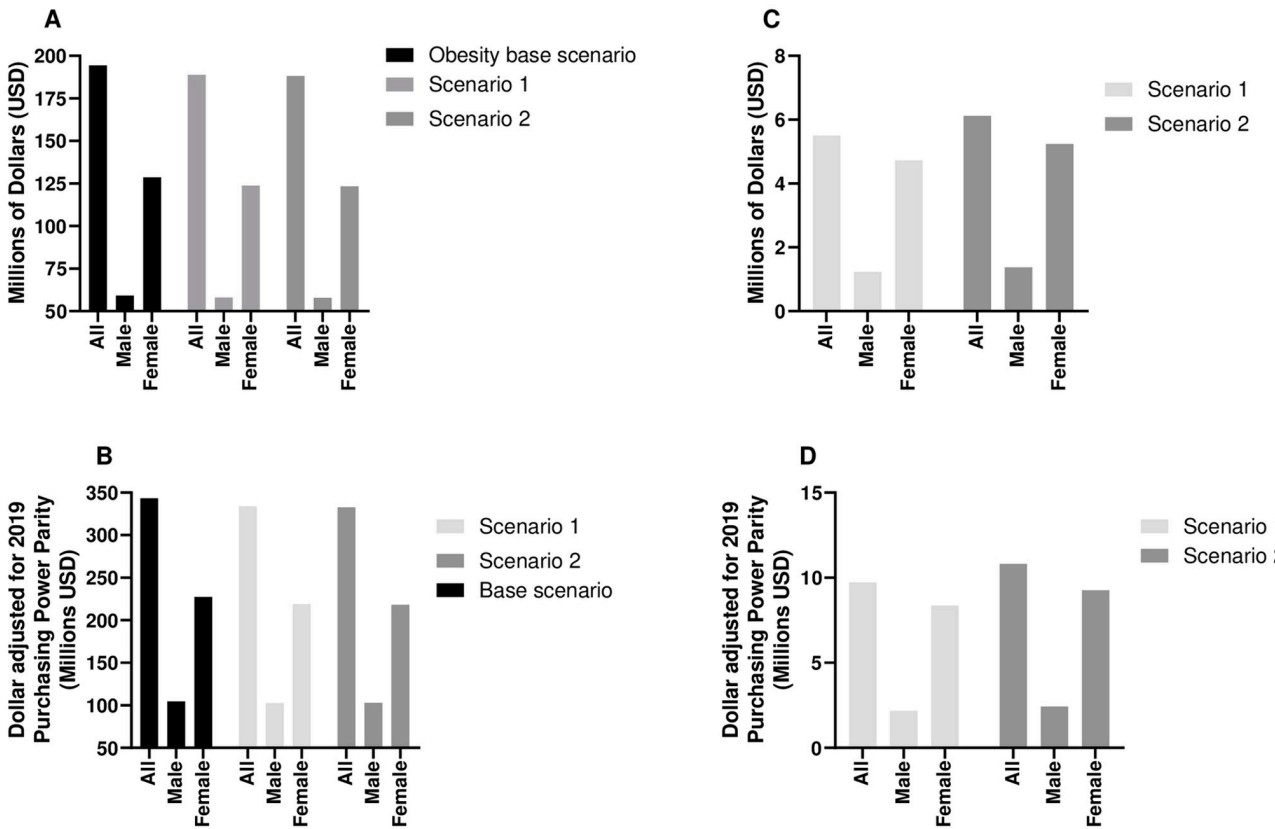

**Fig 2. Direct costs related to obesity in Brazilian adults aged 20 to 59 years and consumers of sugary beverages, and costs that can be avoided five years after FoPNL implementation.** A: Estimation of the direct costs, in US dollars, of public health service related to obesity in Brazilian consumers of sugary beverages aged between 20 and 59 years in different study scenarios and for both sexes. B: Estimation of the direct costs of public health service related to obesity in consumers of sugary beverages, adjusted according to purchasing power parity, in different study scenarios and for both sexes. C: Direct public health costs, in US dollars, associated with obesity that could be avoided in different scenarios. D: Direct costs of public health service related to obesity that could be avoided in different scenarios, adjusted according to purchasing power parity.

prevalence in the two simulated scenarios is expected to reduce the development of chronic non-communicable diseases. Therefore, further studies including this assessment are necessary. A modeling study carried out in Mexico estimated that the implementation of FoPNL could reduce the prevalence of obesity by 2.92% (95% CI, −3.67 to −2.16) due to a reduction in energy consumption from sugary beverages by −23.2 kcal/day/person (95% CI, −24.5 to −21.9) [26]. The other modeling study estimated the effect of FoPNL using various models on mortality from chronic diseases using the Preventable Risk Integrated ModEl (PRIME) macro-simulation in France. The results showed that the FoPNL Nutri Score model estimated a reduction of approximately 3.4% in deaths from food-related chronic non-communicable diseases, mainly cardiovascular diseases [47].

The public health system can also benefit from reduction in the incidence of obesity. The National Health Survey conducted in Brazil showed that more than half of Brazilians (71.5%) depend on the public health system for treatment [39]. Given the current global context of COVID-19 pandemic, the number of Brazilians who depend on the public health system may be even greater, reaching approximately 80% [48]. In this study, it was estimated that the decrease in excess body weight prevalence could lead to a reduction of nearly US$ 5,5 millions (95% CI 4,7 to 8,8) in direct costs related to its treatment within five years of FoPNL implementation. Basto-Abreu et al. [26] estimated a reduction of approximately US$ 1.1 billion in

direct costs related to the treatment of obesity five years after FoPNL implementation in Mexico. In the present study, the rising trend in obesity prevalence in Brazil was considered. This increase can impact the public health system considering the growing need for healthcare. Wang et al. estimated an increase in the prevalence of obesity with a consequent increase in healthcare costs in China [49].

To the best of our knowledge, this is the first study to estimate the impact of FoPNL on the Brazilian population related to the consumption of sugary beverages. Although this study contributes to the insights into the effects of FoPNL, it still has limitations that merit discussion. The simulation was based on the decline in the purchase of all beverages after the first phase of implementation of policies (implementation of FoPNL, restriction marketing, and sales in schools) in Chile, from a different age group in this study. The octagon warning of "high in" was adopted in Chile. However, Chilean population, whose results may be different for the Brazilian population, which still lacks experimental data in this regard. In addition, the differences in both FoPNL systems and public health policies related to sugary beverages may lead to different choices, and consequently, differently impact the energy intake of consumers. However, this research has strengths, including adjustments to make the study more realistic, such as delimitation of the age group between 20 and 59 years, considering a trend time on the consumption of sugary beverages already observed in a previous study [7], and considering global rise in the prevalence of obesity and excess body weight [50].

In addition, although other factors may interfere with consumer choices, only the effect of FoPNL on the energy intake of sugary beverages was simulated in this study. The intake of sugary beverages was evaluated through a survey carried out in Brazilian capitals; therefore, it may not represent the entire population. Another limitation was the estimation of the number of sugary beverages consumed without investigating the volume consumed per can of soda or glass of juice. To address this limitation, the average of the volumes was calculated. Furthermore, another limitation refers to the use of soda as the only sugary beverage in this study. Other sugary beverage may contain fats in their composition, which were not considered in this analysis, which could affect the results obtained in the modeling. Finally, the study considered physical exercise and intake of other nutrients such as fats as constant factors, which can alter the energy balance of an individual [26, 51] and are considered in other types of modeling.

In conclusion, a reduction in energy intake is estimated along with a consequent reduction in the prevalence of obesity by 0.32 percentage points after five years of FoPNL implementation in Brazil. This reduction in obesity prevalence may save US$ 5,5 million (95% CI 4,7 to 8,8) in the public budget. This amount may even be invested in the promotion of physical activity, educational advertisements, and actions aimed at schools. Implementing FoPNL can improve consumer understanding of the nutritional value of a product, making product choice easier. FoPNL has the potential to collaborate in the treatment of obesity, although further actions are necessary to prevent the occurrence of obesity/excess body weight and thus reduce related public health costs.

## Supporting information

**S1 File. Supporting information for "Impact of implementation of front-of-package nutrition labeling on sugary beverage consumption and consequently on the prevalence of excess body weight and obesity and related direct costs in Brazil: An estimate through a modeling study".**
(DOCX)

**S1 Fig. Front-of-package labeling with black magnifying glass design adopted by the Brazilian legislation, according to the Normative Instruction Nº 75 of 2020.** Statements "high

in" from left to right: added sugar; sodium; saturated fat; added sugar, saturated fat and sodium in the same product.
(TIF)

**S2 Fig. Front-of-package nutrition labeling in a warning model and black octagon design adopted by Chilean legislation number 20,606.** From left to right: high in sugars, high in saturated fats, high in sodium, high in calories.
(TIF)

**S3 Fig. Proportion of Brazilians who lived in the Brazilian state capitals and consumed sugary beverages between the years of 2007 and 2019 (expect 2017) and its projection up to 2024.**
(TIF)

**S4 Fig. Average consumption of energy from sugary beverages by Brazilians who lived in the Brazilian state capitals and consumed such beverages between the years of 2007 to 2019 (except 2017) and its projection up to 2024.**
(TIF)

**S5 Fig. Average consumption of sodium from sugary beverages by Brazilians who lived in the Brazilian state capitals and consumed such beverages between the years of 2007 to 2019 (except 2017) and its projection up to 2024.**
(TIF)

**S6 Fig. Front-of-package nutrition labeling in red warning circle design model used in the experimental market study by Acton et al. (2019).**
(TIF)

**S7 Fig. Proportion of Brazilians who consume sugary beverages and have excess body weight living in Brazilian state capitals between the years of 2007 and 2019 (except 2017) and its projection up to the year 2024.**
(TIF)

**S8 Fig. Proportion of Brazilians who consume sugary beverages and are obese living in Brazilian state capitals between the years of 2007 and 2019 (except 2017) and its projection up to the year 2024.**
(TIF)

**S1 Table. Variables acquired from the VIGITEL database and used in the study.**
(DOCX)

**S2 Table. Profile of nutrients and their limits for liquid foods adopted by the Brazilian legislation.**
(DOCX)

**S3 Table. Profile of nutrients and their limits for liquid foods adopted by the Chilean legislation according to each implementation phase.**
(DOCX)

**S4 Table. Beverages included in the study by Taillie et al. (2021).**
(DOCX)

**S5 Table. Average differences in the purchase of beverages, in energy and sodium, before and after implementation of the policies observed by Taillie et al. (2021).**
(DOCX)

**S6 Table. Nutritional composition of beverages (median, interquartile range and average percentage changes) regarding the contents of energy and sodium during the pre-implementation period of the Chilean legislation observed by Kanter et al., (2019).**
(DOCX)

**S7 Table. Purchase of beverages observed in the experimental market study by Acton et al., (2019), in calorie and sodium average and percentage average variation, between the intervention and control groups.**
(DOCX)

**S8 Table. Estimations of Brazilian who consume sugary beverages and the prevalence of excess body weight and obesity up to 2024 based on the temporal trends and projections according to the VIGITEL survey.**
(DOCX)

**S9 Table. Estimations of the prevalence of excess body weight and obesity in Brazilian consumers of sugary beverages after the implementation of front-of-package labeling up to 2024, and sensitivity analysis.**
(DOCX)

**S10 Table. Estimations of the reduction in prevalence of excess body weight and obesity in the Brazilian population attributed to the implementation of the front-of-package up to 2024, and sensitivity analysis.**
(DOCX)

**S11 Table. Estimations of the direct costs regarding obesity according to what Nilson et al. described for the total Brazilian population and for the age group of 20 to 59 years.**
(DOCX)

## Author Contributions

**Conceptualization:** Lucilene Rezende Anastácio.

**Data curation:** Natália Cristina de Faria, Gabriel Machado de Paula Andrade, Luíza Vargas Mascarenhas Braga, Eduardo Augusto Fernandes Nilson, Lucilene Rezende Anastácio.

**Formal analysis:** Natália Cristina de Faria, Gabriel Machado de Paula Andrade, Cristina Mariano Ruas, Eduardo Augusto Fernandes Nilson, Lucilene Rezende Anastácio.

**Funding acquisition:** Lucilene Rezende Anastácio.

**Investigation:** Natália Cristina de Faria, Luíza Vargas Mascarenhas Braga.

**Methodology:** Natália Cristina de Faria, Cristina Mariano Ruas, Eduardo Augusto Fernandes Nilson, Lucilene Rezende Anastácio.

**Project administration:** Lucilene Rezende Anastácio.

**Supervision:** Lucilene Rezende Anastácio.

**Validation:** Cristina Mariano Ruas, Rafael Moreira Claro, Lucilene Rezende Anastácio.

**Visualization:** Natália Cristina de Faria, Lucilene Rezende Anastácio.

**Writing – original draft:** Natália Cristina de Faria, Lucilene Rezende Anastácio.

**Writing – review & editing:** Natália Cristina de Faria, Cristina Mariano Ruas, Rafael Moreira Claro, Lucilene Rezende Anastácio.

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
