## [Decision Letter · Decision Letter 0]

19 Dec 2022

PONE-D-22-26938Impact of the implementation of front-of-package nutrition labeling on sugary beverage consumption on overweight and obesity and related direct costs in Brazil: a modeling studyPLOS ONE

Dear Dr. Rezende,

Thank you for submitting your manuscript to PLOS ONE. After careful consideration, we feel that it has merit but does not fully meet PLOS ONE’s publication criteria as it currently stands. Therefore, we invite you to submit a revised version of the manuscript that addresses the points raised during the review process.

We look forward to receiving your revised manuscript.

Kind regards,

Anselm J. M. Hennis

Academic Editor

PLOS ONE

Journal Requirements:

Additional Editor Comments (if provided):

Dear Authors,

this paper on 'Impact of the implementation of front-of-package nutrition labeling on sugary beverage consumption on overweight and obesity and related direct costs in Brazil: a modeling study', has the potential to contribute to an important area of public health, given the challenge of tackling the obesity epidemic. There are, however, several issues that must be addressed by the authors which have been noted by the Reviewers.

The title needs to clarify that the Impact is a 'projection or an estimate', rather than an evaluation of an established intervention.

There appear to be concerns noted by the authors regarding data access and this is potentially a cause for concern. Additionally, the Ethics approval dates from 2013, and it would be helpful to provide more up-to-date approval.

The Abstract does not coherently present the rationale, methods, results nor conclusion of the study, and like the rest of the paper, would benefit from an editorial revision by an expert English-speaker. This has been noted by Reviewer 1. The Introduction is also verbose and needs to be precise, succinct and focused. The justification for this study as presented in lines 63-65 is weak, and at odds with the subsequent text of lines 85-87.

The term 'direct costs' is not reviewed in the Background, nor is the justification for this evaluation made clear by the Authors. Appropriate information must be provided, and the specific Methods used to calculate such costs, must be clearly presented with supporting justification in the relevant section.

There is also no discussion about the relevance, impact or calculation of the impact of Beverage reformulation in the Introduction, yet this is also a key indicator. Furthermore, the evaluation of the impact of FoPNL is combined with Beverage reformulation in the analyses without clear justification, nor presentation of the specific analyses used.

Statements such as 'Estimated from the perspective of the public health system (line 112) must be clearly explained.

The Methodology used to Estimate energy intake based on use of Vigitel data between 2007 and 2019 (except for 2017) is not clear. The use of linear regression gives the sense that relevant statistical analyses were used, but the Methodology must be presented (referenced relevant formulae) and what it actually means. The lack of data for 2017 must also be explained and its implications for the subsequent estimates.

The meaning of 'soft drinks or artificial juices' needs to be explained in the national context, so the reader can understand the actual beverages captured by this term. Has there been any independent validation of this indicator?

Line 157 speaks to 'The amount of energy (kcal/day) from sugary drinks was calculated based on the nutritional table [29], using standardized procedures'. Please explain these 'standardized procedures'...

The sections on 'Estimation of energy intake reduction, Estimated reduction in body weight, BMI, and prevalence of obesity and overweight, Reduction in obesity and overweight cases, Cost estimation - each need to clearly present the Methods, Formulae used, as well as Justification for the analytical methods used, in a comprehensive and transparent way, that allows the non-expert reader to follow the process. The Methods must be understandable, and this Section must be improved. 

Line 254: Do these Methods consider the appreciation of the real value of money over time? Do the estimates based on the work of Oliveira et al (2011) still have relevance in 2022?.

The Authors also speak to cost savings of $3.5 million over 5 years (line 507); is this the total savings nationally, in the public service or in other sectors??

The Discussion uses 9 pages of text and can be considerably shortened if the Authors present their case precisely and succinctly.

Please note the issues highlighted particularly by Referee 1. 

Reviewers' comments:

Reviewer's Responses to Questions

**Comments to the Author**

1. Is the manuscript technically sound, and do the data support the conclusions?

Reviewer #1: Partly

Reviewer #2: Yes

2. Has the statistical analysis been performed appropriately and rigorously? 

Reviewer #1: Yes

Reviewer #2: I Don't Know

3. Have the authors made all data underlying the findings in their manuscript fully available?

Reviewer #1: Yes

Reviewer #2: Yes

4. Is the manuscript presented in an intelligible fashion and written in standard English?

Reviewer #1: No

Reviewer #2: Yes

5. Review Comments to the Author

Reviewer #1: Overall comments: The paper gives useful information although it is a descriptive level analysis that is presented. There is too much repetition throughout the paper that needs to be addressed, and it would benefit from giving more information about the actual models used than just referencing the models that the simulations were based on. Overall, there is merit to the study. While the grammar in the overall paper is acceptable, the abstract needs proofreading.

The abstract needs reworking, considering the below:

Line 2 should be more impactful as using ‘may” hints at uncertainty. “Intake of sugary beverages has been associated with CNCDS…”

The abstract and main body of the paper do not align in terms of whether the FoPNL has actually been implemented in Brazil. Clarity is needed and changes to with the abstract or main paper made to align.

Abstract needs to be proofread for grammar, for example in the abstract line 7 “A simulation study...” line 10 “The following scenarios…” line 14 Change in body weight was estimated by a simulation model conducted by Hall etc al.” etc.

The scenarios in the abstract are not clear as to what is actually being simulated—it is clearer in the paper methods (line 99) what the three scenarios are.

What does line 15 “Linear trend…” mean in terms of the simulation?

Main paper comments:

Need references for line 63-65.

Line 82, give the direction of the impact on the number of obese people in the country.

Line 90, specify direct costs to whom (i.e. to the public heath system)

Describe what direct costs to the public health system are.

In the scenarios detailed from line 99, what is the definition of reformulation for the purposes of this study which is considered in scenario 3? The literature reviewed in the introduction do not speak to the impact of reformulation and it should be included if it is being used in scenario 3. This can be addressed by moving sections of the discussion to the introduction

What was the reason for excluding persons with extreme BMI (line 134)?

Why was the year 2017 excluded (line 164)?

Line 240 is unclear “…including the one…” –so were there multiple studies and not just one study? The study needs a bit more description for clarity.

Lines 246 to 252 have repetition regarding how obese individuals were calculated, but also does not detail exactly how this subset was determined.

I don not think lines 288-294 or 316-322 or 335-341 or 362-367 are necessary under the tables/figures as they are repeating what has already been described in the methods. Instead the word count saved by taking those lines out can be used to describe in more detail the simulation models that were used as the variables being put into those models were described, but the actual models were referenced.

The discussion section needs to be revised, consider the below comments:

Lines 382-406 are better suited for the introduction.

Line 408-412 is unclear, in the introduction it may help to give more details about the Acton study since it is mentioned repeatedly in this paper.

Again, lines 414-26 should be in the intro, with line 427-428 then referenced as appropriate within the discussion.

Lines 463-474 also could be part of the introduction.

How does the sentence in lines 484-486 related to the findings of your study? If a different model is adopted in Brazil would better outcomes be expected?

Line 507, what percentage of the costs to public health is US$3.5 million? Is the reduction in cost significant?

Again, much of the discussion past line 511 could be in the introduction--the discussion therefore needs to be revisited to determine what should be in the introduction, as well as for repetition.

The overall impact of the simulation is not clear. Is the reduction in obesity impactful as compared to other simulation studies? Is the reduction in cost impactful compared to other studies? While some studies have been referenced, the utility of this study in comparison has not been adequately discussed.

Reviewer #2: 179-181: I don't understand this passage. Is this downward trend in Brazil related to the use of FOPNLs? How is this result connected to the one we see in Canada after using a FOPNL?

Conclusion: I honestly don't see how a reduction of 26 kcal/die and a shift from a BMI of 26.5 kg/m2 to 26.1 kg/m2 in 5 years, with a weight reduction of 1 kg, can be seen as a "potential to reduce overweight and obesity prevalence". Honestly, it also seems to me too big a leap to go from these results to saying that FOPNLs may lead to a reduction in NCDs.

Even if it's true that the public health system can benefit from the reduction in obesity cases, I don't think we can say that FOPNLs alone can achieve that. And since this kind of studies will be used from the policymakers in the near future to implement health policies, I believe it is necessary to always dedicate a relevant section of the papers to the need to implement education policies at a national level to push people to make better choices all round, and not just by comparing (assuming they do) labels at the supermarket.

6. PLOS authors have the option to publish the peer review history of their article (what does this mean?). If published, this will include your full peer review and any attached files.

Reviewer #1: No

Reviewer #2: No

---

## [Author Response · Author response to Decision Letter 0]

21 Mar 2023

Dear Authors,

This paper on 'Impact of the implementation of front-of-package nutrition labeling on sugary beverage consumption on overweight and obesity and related direct costs in Brazil: a modeling study' has the potential to contribute to an important area of public health, given the challenge of tackling the obesity epidemic. There are, however, several issues that must be addressed by the authors which have been noted by the Reviewers.

Dear Editor,

We appreciate the opportunity to review and improve our article. We would like to thank the reviewers for their time and essential comments. In this review, we address the reviewers' suggestions and the important comments of the Ph.D. qualifying exam committee of the Ph.D. student, Natália C. Faria. They have suggested improvements in the experimental data - as we now have real evidence of FoPNL on purchase from Chile (and this is also a closer population about Brazil than Canadians). So, we have also changed the experimental source of data too.

The title needs to clarify that the Impact is a 'projection or an estimate', rather than an evaluation of an established intervention.

We rewrote the title, including your suggestion: Impact of the implementation of front-of-package nutrition labeling on sugary beverage consumption on excess body weight and obesity and related direct costs in Brazil: an estimate through a modeling study.

There appear to be concerns noted by the authors regarding data access and this is potentially a cause for concern. 

We agreed and removed the part of the data not shown and adjusted the work method. We entered these data as part of the sensitivity analysis described in the work method and with a detailed table in a supplementary file.

Line 291: “Sensitivity analysis:

Line 308: The range of uncertainty regarding the impact of FoPNL implementation on the prevalence of obesity and excess body weight was determined using sensitivity analysis. For this purpose, alternative scenarios were simulated. The scenario 3 impact promoted by FoPNL in adults (n = 1,213) regarding the purchase of sweetened beverages observed in the Canadian experimental market study (energy: −10.5%; sodium: −57.56%) was used [40,41], which was consistent with the trend of consumption of sugary beverages in the Brazilian population. Acton et al. obtained their results through an experimental market study conducted in Canada using a “high in” FoPNL in a red circle [40]. Finally, scenario 4 was based on reformulation of beverages (energy: −1.6%; sodium: 1.8%) observed by Kanter et al. in Chile [14], and was considered to be associated with scenario 3 described above.

Additionally, the Ethics approval dates from 2013, and it would be helpful to provide more up-to-date approval.

Dear Editor, we appreciate your comment. The survey VIGITEL was approved by the National Commission of Ethics in Research for Human Beings of the Ministry of Health (CONEP Opinion 355.590 of 26 June 2013 and certificate of presentation for ethics assessment - CAAE - number 16202813.2.0000.0008). It is mentioned as it is in many other papers that used this database, like:

https://doi.org/10.1590/1516-3180.2017.0044250517

https://doi.org/10.1186/s12939-021-01533-z

https://doi.org/10.1007/s10433-021-00659-x

https://doi.org/10.1590/S1679-49742021000100008

The Abstract does not coherently present the rationale, methods, results nor conclusion of the study, and like the rest of the paper, would benefit from an editorial revision by an expert English-speaker. This has been noted by Reviewer 1. 

We appreciate the editor's comments about our paper. We have now included the sections mentioned on the Abstract. Our paper has passed through a native English speaker review. 

The Introduction is also verbose and needs to be precise, succinct and focused. The justification for this study as presented in lines 63-65 is weak, and at odds with the subsequent text of lines 85-87.

We thank the editor for this important observation. We have included some important elements that reviewer 2 asked us for and removed some text to improve the justification and precision. We also tried to improve the justification of the study.

The term 'direct costs' is not reviewed in the Background, nor is the justification for this evaluation made clear by the Authors. Appropriate information must be provided, and the specific Methods used to calculate such costs, must be clearly presented with supporting justification in the relevant section.

We appreciate the important observation and have included the term direct costs in the background and justification of the study. We also created a supplementary file to explain in more detail the methods developed in work.

In the Abstract, line 33: “Rationale: Intake of sugary beverages has been associated with obesity and chronic non-communicable diseases, thereby increasing the direct health costs related to these diseases.”

And in the introduction, in line 80: “Obesity and its effects on health generated an estimate annual costs of R$ 1.42 billion (95% CI, 0.98–1.87) in 2018 via the Public Health System (SUS) in Brazil [8]. The costs included hospitalizations, outpatient procedures, and medications distributed by the SUS for the treatment of these diseases, excluding supplementary health costs in the country, as well as the economic and social costs associated with illness and death from these causes [8]. Direct costs are those paid by health services related to immediate expenses, and include labor, tests, and medications [9].”

There is also no discussion about the relevance, impact or calculation of the impact of Beverage reformulation in the Introduction, yet this is also a key indicator. Furthermore, the evaluation of the impact of FoPNL is combined with Beverage reformulation in the analyses without clear justification, nor presentation of the specific analyses used.

We thank you and agree with the suggestion. The relevance of reformulating beverages was included in the introduction, as shown in the excerpt below. In the supplementary file we show more details of the analyzes used.

In the Introduction, line 99: “In Chile, the food industry reformulated foods and beverages after implementing FoPNL and other public health policies, reducing added sugars in sugary beverages and introducing non-nutritive sweeteners [14,15]. Kanter et al. found a reduction in the content of added sugars in beverages between 2015 and 2016 (median 7.5 g/100 mL, interquartile range (IQR): 2.3–10.0 in 2015 to 6.0 g/100 mL, IQR: 2.2–10 in 2016) in Chile, referring to reformulation [14].” 

Statements such as 'Estimated from the perspective of the public health system (line 112) must be clearly explained.

We appreciate the suggestion and expand on the explanation as described below:

In the Method, line 154: “Additionally, the reduction in direct costs related to obesity and excess body weight in the Brazilian population was estimated from the perspective of the public health system. In Brazil, a significant portion of the population depends on SUS, representing 71.5% of Brazilians who do not have access to supplementary health services [31]. Most SUS expenses related to disease treatment are based on outpatient procedures, hospital care, and dispensing of medicines to control chronic diseases by the Popular Pharmacy Program [32].”

The Methodology used to Estimate energy intake based on use of Vigitel data between 2007 and 2019 (except for 2017) is not clear. The use of linear regression gives the sense that relevant statistical analyses were used, but the Methodology must be presented (referenced relevant formulae) and what it actually means. The lack of data for 2017 must also be explained as its implications for the subsequent estimates.

We agree and have included explanations in the method of work and supplementary file. In calculating the tendency to consume sugary beverages in the Brazilian population, database from 12 years reported by individuals Brazilians in the VIGITEL survey were analyzed. However, the question analyzed for calculating the consumption trend in the Vigitel survey differed in 2017 compared to the other years analyzed. In 2017, the question referred to the amount of consumption of sugary beverages on the day before the survey, while in other years, it was related to the usual amount when consuming sugary beverages. As they are different questions, the year 2017 was excluded. This explanation was included in the supplementary material developed.

In the Method, line 285: “The analyzed population was described according to characteristics related to sex and age and presented as the mean and 95% confidence interval (95% CI). The prevalence of sugary beverage consumers and their average daily consumption, stratified by sex, were estimated for each year using the VIGITEL survey. The consumption of sugary beverages was presented as the average and 95% CI. Linear trends were investigated using linear regression analysis. The dependent variables evaluated for each year included the annual proportion of Brazilians who consumed sugary beverages, average amount of sugary beverages consumed (in terms of energy and sodium), annual proportion of obese Brazilians who consumed sugary beverages, and annual proportion of Brazilians with excess body weight who consumed sugary beverages. The independent variables included data of the VIGITEL survey or all years used in this study. Regression models were stratified by sex, considering the sample weights of the VIGITEL survey. Thus, temporal trends (2007–2019, except 2017) were analyzed, and the projected prevalence (2020–2024) was calculated in the base scenario. Additional details are provided in the supplementary file.”

The meaning of 'soft drinks or artificial juices' needs to be explained in the national context, so the reader can understand the actual beverages captured by this term. Has there been any independent validation of this indicator?

We thank the reviewer for the important observation. A definition of soft drinks and artificial juices according brazilian regulations were included in the text. Line 196: “According to the Technical Regulation in Brazil, soft drinks are carbonated beverages obtained by dissolving the juice or plant extract in drinking water, added sugar, and saturated with industrially pure carbon dioxide [35]. In this study, only soft drinks were considered as sugary beverages.” 

There was a validation and reproducibility analysis carried out in random sub-samples (n=112 and n=109) of the total number of participants (N=2,204) in adults aged 18 years or older, in 2005, of the VIGITEL survey. Among the evaluated indicators were the evaluation of the daily or almost daily consumption of soft drinks. For reproducibility analysis, the results obtained in the original telephone interview were compared with results obtained in another telephone interview carried out between seven and 15 days after the original interview. Kappa coefficients indicated substantial agreement (all: 0.77; men: 0.83; women: 0.72). For validity analysis, the results of the telephone interview were compared with three 24-hour recalls carried out 15 days after the original interview. Reasonable values of specificity (all: 94.1%; men: 93.3%; women: 94.6%) and sensitivity (all: 87.5%; men: 50%; women: 100%) were observed for soft drink consumption.

Details are described in this publication: 

https://doi.org/10.1590/S0034-89102008000400002

We included this information in the method (line 264) and supplementary file.

Line 193: “The questions related to soft drink consumption used in the VIGITEL survey passed through a previous validation analysis with reasonable values of specificity (all: 94.1%) and sensitivity (all: 87.5%) [34].”

Although this is an important question, all the sugary beverage consumption was considered as “soft drink” consumption based on the nutritional composition of this beverage.

Line 157 speaks to 'The amount of energy (kcal/day) from sugary drinks was calculated based on the nutritional table [29], using standardized procedures'. Please explain these 'standardized procedures'...

We appreciate the suggestion and add in these procedures to a method.

In the Method, line 200: The average consumption of sugary beverages (mL/day) was estimated conform described in a previous study [7]. The average consumption volume was defined as 250 mL to represent the average value between a glass (150 mL) and a can (350 mL) of soda based on the VIGITEL data. The amount of calorie (kcal/day) intake from sugary beverages was based on the Brazilian Table of Food Composition [36]. Further details can be found in the supplementary file.”

The sections on 'Estimation of energy intake reduction, Estimated reduction in body weight, BMI, and prevalence of obesity and overweight, Reduction in obesity and overweight cases, Cost estimation - each need to clearly present the Methods, Formulae used, as well as Justification for the analytical methods used, in a comprehensive and transparent way, that allows the non-expert reader to follow the process. The Methods must be understandable, and this Section must be improved. 

We appreciate the suggestion and have created a supplemental file to detail the methods used.

Line 254: Do these Methods consider the appreciation of the real value of money over time? Do the estimates based on the work of Oliveira et al (2011) still have relevance in 2022?.

We appreciate the suggestion and decided to change the cost baseline study. We replaced it with the study by Nilson et al., 2020 (https://doi.org/10.26633/RPSP.2020.32). This study describes the direct costs attributable to obesity, hypertension and diabetes in Brazil's unified health system (SUS) in the year (2018). Thus, the value update referred to only one year.

The Authors also speak to cost savings of $3.5 million over 5 years (line 507); is this the total savings nationally, in the public service or in other sectors??

This possible economy will be at the national level, for the single health system in Brazil. We made it clearer in line 398.

The Discussion uses 9 pages of text and can be considerably shortened if the Authors present their case precisely and succinctly.

Please note the issues highlighted particularly by Referee 1. 

We agree and completely reframe the discussion.

Reviewers' comments:

Reviewer's Responses to Questions

Comments to the Author

1. Is the manuscript technically sound, and do the data support the conclusions?

Reviewer #1: Partly

Reviewer #2: Yes

2. Has the statistical analysis been performed appropriately and rigorously?

Reviewer #1: Yes

Reviewer #2: I Don't Know

3. Have the authors made all data underlying the findings in their manuscript fully available?

Reviewer #1: Yes

Reviewer #2: Yes

4. Is the manuscript presented in an intelligible fashion and written in standard English?

Reviewer #1: No

Reviewer #2: Yes

5. Review Comments to the Author

Reviewer #1: Overall comments: The paper gives useful information although it is a descriptive level analysis that is presented. There is too much repetition throughout the paper that needs to be addressed, and it would benefit from giving more information about the actual models used than just referencing the models that the simulations were based on. Overall, there is merit to the study. While the grammar in the overall paper is acceptable, the abstract needs proofreading.

We appreciate the opportunity and have reviewed the entire article. We have created a supplemental file to detail the methods of the work.

The abstract needs reworking, considering the below:

Line 2 should be more impactful as using ‘may” hints at uncertainty. “Intake of sugary beverages has been associated with CNCDS…”

We agree and adjust the text.

The abstract and main body of the paper do not align in terms of whether the FoPNL has actually been implemented in Brazil. Clarity is needed and changes to with the abstract or main paper made to align.

We apologize for this and have revised the text. The legislation passed in October 9th, 2020 and implementation of FoPNL in Brazil started on October 9th, 2022.

Abstract needs to be proofread for grammar, for example in the abstract line 7 “A simulation study...” line 10 “The following scenarios…” line 14 Change in body weight was estimated by a simulation model conducted by Hall etc al.” etc.

We apologize and pass all work through a review.

The scenarios in the abstract are not clear as to what is actually being simulated—it is clearer in the paper methods (line 99) what the three scenarios are.

We are sorry for that. Now, the scenarios in the abstract are described as (line 43):

“The following scenarios were considered: base (trend in sugary beverage intake); 1 (base scenario associated with the changes in energy content of the purchased beverages observed after the first phase of the Chilean labeling law (−9.9%); and 2 (scenario 1 associated with reformulation of beverages, total energy reduction of −1.6%).”

What does line 15 “Linear trend…” mean in terms of the simulation?

Linear regression was used to analyze the temporal trend prevalence of obesity/excess body weight in Brazilian population. We adjust the text:

In abstract, line 48: “A linear trend in the prevalence of obesity and excess body weight in the Brazilian population was considered.” 

Main paper comments:

Need references for line 63-65.

We appreciate your note and have included the references (line 177):

doi:10.3389/fnut.2022.898021

doi:10.3389/fnut.2022.921065

doi:10.1371/journal.pone.0265990

Line 82, give the direction of the impact on the number of obese people in the country.

We agree with the suggestion and adjust the text, as follows:

Line 118: “A modeling study conducted in Mexico estimated that FoPNL can promote a reduction of 23.2 kcal/day (95% CI, −24.5 to −21.9) associated with the consumption of sugary drinks and 13.6 kcal/day (95% CI, −14.1 to −13.1) to snacks, resulting in 4.98pp reduction in the number of obese people in the country [26].”

Line 90, specify direct costs to whom (i.e. to the public health system).

We appreciate the suggestion and have included it in the text, as described below:

Line 131: “Thus, the objective of this study was to estimate, over five years, the reduction in the prevalence of obesity and excess body weight among Brazilian adults and the direct costs in public health system related to such problems after the implementation of FoPNL in Brazil.”

Describe what direct costs to the public health system are.

We complete the text, as below, in the introduction::

Line 80: “Obesity and its effects on health generated an estimate annual costs of R$ 1.42 billion (95% CI, 0.98–1.87) in 2018 via the Public Health System (SUS) in Brazil [8]. The costs included hospitalizations, outpatient procedures, and medications distributed by the SUS for the treatment of these diseases, excluding supplementary health costs in the country, as well as the economic and social costs associated with illness and death from these causes [8]. Direct costs are those paid by health services related to immediate expenses, and include labor, tests, and medications [9].

In the scenarios detailed from line 99, what is the definition of reformulation for the purposes of this study which is considered in scenario 3? The literature reviewed in the introduction do not speak to the impact of reformulation and it should be included if it is being used in scenario 3. This can be addressed by moving sections of the discussion to the introduction

We appreciate the important note and have included the excerpt below about the recast in the introduction.

Line 99: “In Chile, the food industry reformulated foods and beverages after implementing FoPNL and other public health policies, reducing added sugars in sugary beverages and introducing non-nutritive sweeteners [14,15]. Kanter et al. found a reduction in the content of added sugars in beverages between 2015 and 2016 (median 7.5 g/100 mL, interquartile range (IQR): 2.3–10.0 in 2015 to 6.0 g/100 mL, IQR: 2.2–10 in 2016) in Chile, referring to reformulation [14].”

What was the reason for excluding persons with extreme BMI (line 134)?

There is a possibility that these people have atypical habits in food consumption, so to standardize we exclude the extremes. The same procedure was performed in another study: 

https://doi.org/10.1371/journal.pmed.1003221

Why was the year 2017 excluded (line 164)?

To calculate the tendency of consumption of sugary beverages in the Brazilian population, data from 12 years reported by Brazilians in the Vigitel survey were analyzed. However, the question analyzed for calculating the consumption trend in the Vigitel survey was different in 2017 compared to the other years analyzed. In 2017, the question refers to the amount of consumption of sugary drinks on the day before the survey, while in other years, it is related to the usual amount when consuming sugary drinks. As they are different questions, the year 2017 was excluded. This explanation was included in the supplementary material developed.

Line 240 is unclear “…including the one…” –so were there multiple studies and not just one study? The study needs a bit more description for clarity.

We have changed the base article used for the cost analysis. The topic Cost estimation in method was rewritten.

Lines 246 to 252 have repetition regarding how obese individuals were calculated, but also does not detail exactly how this subset was determined.

The estimate of the number of obesity cases was obtained by the difference between scenarios 1 and 2 with the temporal trend of obesity prevalence. We have added more information in the supplementary material. 

I don not think lines 288-294 or 316-322 or 335-341 or 362-367 are necessary under the tables/figures as they are repeating what has already been described in the methods. Instead the word count saved by taking those lines out can be used to describe in more detail the simulation models that were used as the variables being put into those models were described, but the actual models were referenced.

We appreciate the suggestion and are removing the suggested text.

The discussion section needs to be revised, consider the below comments:

Lines 382-406 are better suited for the introduction.

We appreciate the suggestion and have completely revamped the discussion.

Line 408-412 is unclear, in the introduction it may help to give more details about the Acton study since it is mentioned repeatedly in this paper.

We chose to remove this section because we changed the base study of the modeling.

Again, lines 414-26 should be in the intro, with line 427-428 then referenced as appropriate within the discussion.

Lines 463-474 also could be part of the introduction.

We appreciate the suggestion and move the text to the introduction section.

How does the sentence in lines 484-486 related to the findings of your study? If a different model is adopted in Brazil would better outcomes be expected?

Evidence on the effect that the black magnifying glass design's effect on the population is still scarce. Studies show that, depending on the FoPNL design adopted, different effects are observed. As the black magnifying glass design was adopted in Brazil, with a different cut-off point for critical nutrients adopted in Chile, the results may be different than expected. However, the scenario based on the Chilean study was adopted because it is an evaluation after the implementation of public policy, in addition to considering some compensation in the consumption of beverages (reduction in purchases of “high em” beverages and increase in purchases of beverages without FoPNL), relevant points for the methodological choice.

We have now included the differences between our labeling system and the Chilean on the supplementary material to a better understanding. 

Line 507, what percentage of the costs to public health is US$3.5 million? Is the reduction in cost significant? 

This reduction represents 2.8% of the total health costs with obesity in the 20-59 age group, adjusted for 2024 (with updated cost estimates based on latest database work – US$5.5 millions).

The overall impact of the simulation is not clear. Is the reduction in obesity impactful as compared to other simulation studies? Is the reduction in cost impactful compared to other studies? While some studies have been referenced, the utility of this study in comparison has not been adequately discussed.

This study shows that the implementation of FoPNL in Brazil has the potential to slow down the increase in the prevalence of obesity and overweight in the Brazilian population that consumes sugary drinks. FoPNL represents one of the public policies that can be used as an obesity prevention strategy. Due to the vast majority of Brazilians being dependent on public health services, the reduction in the prevalence of obesity and, consequently, of CNCDs has the potential to reduce the burden on the system and thus impacts on the reduction of its costs. Bastos-Abreu et al. (2020) also projected the impact of FoPNL implementation in the Mexican population. However, it did not show a temporal trend in the prevalence of obesity. Also, this study and Bastos-Abreu et al. corroborate the results of the reduction of cases of obesity.

Reviewer #2: 179-181: I don't understand this passage. Is this downward trend in Brazil related to the use of FOPNLs? How is this result connected to the one we see in Canada after using a FOPNL?

We reviewed the study and considered the study by Taillie et al. (2021), which evaluated the effect after the first phase of the implementation of Chilean legislation. As there is some compensation in the consumption of beverages and because the policy is being implemented, we believe the results can be closer to what can happen in Brazil.

Conclusion: I honestly don't see how a reduction of 26 kcal/die and a shift from a BMI of 26.5 kg/m2 to 26.1 kg/m2 in 5 years, with a weight reduction of 1 kg, can be seen as a "potential to reduce overweight and obesity prevalence". Honestly, it also seems to me too big a leap to go from these results to saying that FOPNLs may lead to a reduction in NCDs.

Dear reviewer, we thank you for your important comment. In fact, simulation studies are done worldwide the way we have performed. We agree and point out that the FoPNL implementation strategy can help to prevent cases of obesity and overweight and slow down your increase, as described in the last paragraph, copied below:

Line 524: “In conclusion, a reduction in energy intake is estimated along with a consequent reduction in the prevalence of obesity by 0.32pp after five years of FoPNL implementation in Brazil. This reduction in obesity prevalence may save US$ 5,5 million (95% CI 4,7 to 8,8) in the public budget. This amount may even be invested in the promotion of physical activity, educational advertisements, and actions aimed at schools. Implementing FoPNL can improve consumer understanding of the nutritional value of a product, making product choice easier. FoPNL has the potential to collaborate in the treatment of obesity, although further actions are necessary to prevent the occurrence of obesity/excess body weight and thus reduce related public health costs.” 

Although the impact of energy reduction is modest, it is worth mentioning that, in this study, the effect of implementing FoPNL was modeled in a single product category, which are sugary beverages (soft drinks). Other studies have modeled modest reductions in energy intake and shown benefits in the population, such as those highlighted below:

https://doi.org/10.1371/journal.pmed.1003221

https://doi.org/10.1016/j.amepre.2019.06.022

https://doi.org/10.1038/ijo.2010.228

https://doi.org/10.1186/s12966-019-0817-2,

This category of beverages, is consumed by almost half of the Brazilian adults residing in Brazilian capitals. In addition, other public health policies can be implemented, such as taxation and restrictions on product marketing, as well as food and nutrition education actions in the population to intensify obesity prevention.

Even if it's true that the public health system can benefit from the reduction in obesity cases, I don't think we can say that FOPNLs alone can achieve that. And since this kind of studies will be used from the policymakers in the near future to implement health policies, I believe it is necessary to always dedicate a relevant section of the papers to the need to implement education policies at a national level to push people to make better choices all round, and not just by comparing (assuming they do) labels at the supermarket.

We agree with this and highlight in the last paragraph the need to implement other strategies to prevent obesity (line 527). In addition, in the discussion, we highlight that in Chile, other actions were developed, such as restrictions on marketing and sales in schools, as well as taxation, to work with the implementation of FoPNL (line 503).

---

## [Decision Letter · Decision Letter 1]

19 Jun 2023

PONE-D-22-26938R1Impact of implementation of front-of-package nutrition labeling on sugary beverage consumption and consequently on the prevalence of excess body weight and obesity and related direct costs in Brazil: An estimate through a modeling studyPLOS ONE

Dear Dr. Rezende 

Thank you for submitting your manuscript to PLOS ONE. After careful consideration, we feel that it has merit but does not fully meet PLOS ONE’s publication criteria as it currently stands. Therefore, we invite you to submit a revised version of the manuscript that addresses the points raised during the review process.

Please submit your revised manuscript by Aug 03 2023 11:59PM. If you will need more time than this to complete your revisions, please reply to this message or contact the journal office at plosone@plos.org. Please include the following items when submitting your revised manuscript:A rebuttal letter that responds to each point raised by the academic editor and reviewer(s). You should upload this letter as a separate file labeled 'Response to Reviewers'.A marked-up copy of your manuscript that highlights changes made to the original version. You should upload this as a separate file labeled 'Revised Manuscript with Track Changes'.An unmarked version of your revised paper without tracked changes. You should upload this as a separate file labeled 'Manuscript'.If applicable, we recommend that you deposit your laboratory protocols in protocols.io to enhance the reproducibility of your results. Protocols.io assigns your protocol its own identifier (DOI) so that it can be cited independently in the future. For instructions see: https://journals.plos.org/plosone/s/submission-guidelines#loc-laboratory-protocols. Additionally, PLOS ONE offers an option for publishing peer-reviewed Lab Protocol articles, which describe protocols hosted on protocols.io. Read more information on sharing protocols at https://plos.org/protocols?utm_medium=editorial-email&utm_source=authorletters&utm_campaign=protocols.

We look forward to receiving your revised manuscript.

Kind regards,

Anselm J. M. Hennis

Academic Editor

PLOS ONE

Journal Requirements:

Additional Editor Comments:

While many of the issues have been addressed, there are a number of editorial issues highlighted by a referee which need to be addressed.

Of note a concern has also been raised about the importance of sensitivity analyses around the estimates made using statistical modeling techniques, and it would be important for the authors to respond to this.

Reviewers' comments:

Reviewer's Responses to Questions

**Comments to the Author**

1. If the authors have adequately addressed your comments raised in a previous round of review and you feel that this manuscript is now acceptable for publication, you may indicate that here to bypass the “Comments to the Author” section, enter your conflict of interest statement in the “Confidential to Editor” section, and submit your "Accept" recommendation.

Reviewer #2: All comments have been addressed

Reviewer #3: (No Response)

2. Is the manuscript technically sound, and do the data support the conclusions?

Reviewer #2: Yes

Reviewer #3: Partly

3. Has the statistical analysis been performed appropriately and rigorously? 

Reviewer #2: I Don't Know

Reviewer #3: I Don't Know

4. Have the authors made all data underlying the findings in their manuscript fully available?

Reviewer #2: Yes

Reviewer #3: Yes

5. Is the manuscript presented in an intelligible fashion and written in standard English?

Reviewer #2: Yes

Reviewer #3: Yes

6. Review Comments to the Author

Reviewer #2: (No Response)

Reviewer #3: Abstract

Line 3: The word nutrition should be inserted before labelling so that the meaning of the abbreviation is clear.

There are several places in abstract where grammar/syntax is compromised presumably to conserve words. This should not be done as it makes the text less clear.

Line 5: To estimate, over five years, THE impact of implementing

Line 7: A Simulation study to

Line 9: The VIGITEL research

Line 9: VIGITEL research database (2019)- do you mean the 2019 version or data for the year 2019? That is unclear.

Line 9: Simply saying “n” is insufficient. Should say sample size if that is what n stands for.

Line 10: Instead of “Following scenarios were considered” could be “The scenarios considered were:”

Line 19: “CI95%” - should be 95% CI

Line 23: the abbreviation “pp” is unclear to me. If it stands for ‘percentage points’ which I am assuming/guessing from the main text then this should be clearer from the abstract.

Introduction

Lines 64-65:

“However, even though the impact of FoPNL on food purchase

64 and consumption in the Brazilian population has been investigated, it still needs

65 further studies.”

Recommendation: Authors could indicate here what further investigations are needed. The statement as stands in vague. Given their study, it could read further investigations needed to understand how consumption has impacted obesity rates etc.

Lines 66-70 references a meta-analysis but not the population/country where done. But country is mentioned for other studies in the same and ensuing paragraphs. Feels inconsistent.

Methods

Generally very well explained.

I found the system of referring to base scenario then scenario 1 and scenario 2 somewhat confusing. Maybe rename so scenario 1 is base scenario or base scenario could be 0.

Line 117- “database for 2019”. Does that mean the version released in 2019 or data for the year 2019?

Line 193: The authors state: The effect of reducing energy intake of sugary drinks from changes in FoPNL disregarded saturated fats and sugars to estimate body weight variation.

Can the authors explain how this might have under or overestimated the overall effects of FOPNL on obesity prevalence?

Major: In modelling studies, given the nature of simulations and predictions it is best practice to run sensitivity analyses around the assumptions made. This is a major omission from this paper. The authors chose Acton et al for estimating the effect of the FOPNL on SS consumption. The reason is sound but given that a meta-analysis exists, the findings from the meta-analysis and/or studies from the analysis should be used as part of a sensitivity analysis in the model to determine the extent of the difference had a different paper/estimate been chosen.

Results

Once again, generally well presented with a few queries:

Table 2 and the associated text line 298-304. The text states that the mean BMI of analysed individuals was 26.5 which I assumed was there BMI at the beginning of the five year modelling exercise. In the Table it states that the BMI in base scenario is 26.5. This confused me because I thought the mean BMI of 26.5 referred to the mean BMI before any scenarios were applied. Thus I am expecting that there should be a figure that estimates the change in BMI under the base scenario where change in consumption is reduced by -20.

In other worse, the base scenario represents a reduction in consumption of -20 after 5 years (Table 1) thus there should also be some reduction in BMI from this just as there is for scenario 1 where the expected change after 5 years is -26. Can the authors please clarify the above?

Line 306- Figures 1: C1, C2 and C3 do not show a reduction in obesity prevalence and so appears to contradict the text. I believe the authors are referring to the reduction in scenario 1 relative to the base scenario. That is not stated in text and the particular graphs being referred to is not clear.

Line 308 refers to Figure 1 but there are six graphs attached to Figure 1. Is the 0.25 to be read from Graph A, B-1 B-2 or the series of C graphs?

Line 308- Can you clarify that ‘pp’ refers to percentage points? It does not appear in the graphs only in text.

The reference in line 310 to numbers of cases of obesity avoided after FOPNL is more intuitive and informative than the percentage points more commonly referred to by authors. I recommend more use of this. Perhaps use this in the abstract to make it easier for policy makers to understand and provide more concrete thinking in terms of the impact of the FOPNL.

Discussion

This was generally very long with information that reached outside the scope of the paper insufficient focus on discussing the limitation such as the potential limitation arising from line 193.

For example, the two pages spent explaining Chile’s journey through all obesity prevention policies could be summarized into one paragraph which focuses on the impact FOPNL and the multiple policies has had on obesity/overweight prevalence since that is the focus of this paper.

7. PLOS authors have the option to publish the peer review history of their article (what does this mean?). If published, this will include your full peer review and any attached files.

Reviewer #2: No

Reviewer #3: No

---

## [Author Response · Author response to Decision Letter 1]

10 Jul 2023

Response to reviewers

Journal Requirements:

We’ve reviewed the references.

Additional Editor Comments:

While many of the issues have been addressed, there are a number of editorial issues highlighted by a referee which need to be addressed.

We appreciate the opportunity to review this paper again, and we answered to all of the questions listed by the reviewer.

We want to emphasize that reviewer #3 probably reviewed the first version we submitted. We wonder if this happened because he/she asked for suggestions we had already addressed. Also, the number of the lines he/she mentioned were not the same as the last reviewed paper. 

Of note a concern has also been raised about the importance of sensitivity analyses around the estimates made using statistical modeling techniques, and it would be important for the authors to respond to this.

We appreciate the reviewer's suggestions and have answered the question about sensitivity analysis below. Sensitivity analysis was included in the first revision.

Reviewers' comments:

Reviewer's Responses to Questions

Comments to the Author

1. If the authors have adequately addressed your comments raised in a previous round of review and you feel that this manuscript is now acceptable for publication, you may indicate that here to bypass the “Comments to the Author” section, enter your conflict of interest statement in the “Confidential to Editor” section, and submit your "Accept" recommendation.

Reviewer #2: All comments have been addressed

Reviewer #3: (No Response)

2. Is the manuscript technically sound, and do the data support the conclusions?

Reviewer #2: Yes

Reviewer #3: Partly

3. Has the statistical analysis been performed appropriately and rigorously?

Reviewer #2: I Don't Know

Reviewer #3: I Don't Know

4. Have the authors made all data underlying the findings in their manuscript fully available? The PLOS Data policy requires authors to make all data underlying the findings described in their manuscript fully available without restriction, with rare exception (please refer to the Data Availability Statement in the manuscript PDF file). The data should be provided as part of the manuscript or its supporting information, or deposited to a public repository. For example, in addition to summary statistics, the data points behind means, medians and variance measures should be available. If there are restrictions on publicly sharing data—e.g. participant privacy or use of data from a third party—those must be specified.

Reviewer #2: Yes

Reviewer #3: Yes

5. Is the manuscript presented in an intelligible fashion and written in standard English? PLOS ONE does not copyedit accepted manuscripts, so the language in submitted articles must be clear, correct, and unambiguous. Any typographical or grammatical errors should be corrected at revision, so please note any specific errors here.

Reviewer #2: Yes

Reviewer #3: Yes

6. Review Comments to the Author

Reviewer #2: (No Response)

Reviewer #3: Abstract

Dear Reviewer, we appreciate your essential contributions to our manuscript and the opportunity to answer to your questions. The version you used to send us suggestions has already gone through a first review. Therefore, we added the responses to the last version of the article. 

Line 3: The word nutrition should be inserted before labelling so that the meaning of the abbreviation is clear.

There are several places in abstract where grammar/syntax is compromised presumably to conserve words. This should not be done as it makes the text less clear.

We included the word “nutrition” in the requested line. Thank you for your observation (line 35). 

Line 5: To estimate, over five years, THE impact of implementing

We thank you and adapted the sentence as suggested (lines 37-38).

Line 7: A Simulation study to

We adapted the text as proposed. Thank you for that (line 40).

Line 9: The VIGITEL research

Line 9: VIGITEL research database (2019)- do you mean the 2019 version or data for the year 2019? That is unclear.

We used data collected in 2019 and published in the 2020 report. As requested, we have adapted the text to be clearer (line 43).

Line 9: Simply saying “n” is insufficient. Should say sample size if that is what n stands for.

We included this explanation for clarity in lines 43-44: (the final sample consisted of 12,471 data points representing 14,380,032 Brazilians).

Line 10: Instead of “Following scenarios were considered” could be “The scenarios considered were:”

We appreciate the suggestion and changed the text accordingly (line 45).

Line 19: “CI95%” - should be 95% CI

We apologize for this mistyping and we have revised all of the text to 95% CI.

Line 23: the abbreviation “pp” is unclear to me. If it stands for ‘percentage points’ which I am assuming/guessing from the main text then this should be clearer from the abstract.

We removed the abbreviation and used “percentage points” instead to improve understanding (lines 60-62).

Introduction

Lines 64-65:

“However, even though the impact of FoPNL on food purchase

64 and consumption in the Brazilian population has been investigated, it still needs

65 further studies.”

Recommendation: Authors could indicate here what further investigations are needed. The statement as stands in vague. Given their study, it could read further investigations needed to understand how consumption has impacted obesity rates etc.

We added the sentence below, on lines 132-135: 

“In addition, investigations are needed to understand consumer behavior because of the implementation of FoPNL, such as possible changes in purchasing patterns, consequent changes in food consumption and causing changes in nutritional status.”

Lines 66-70 references a meta-analysis but not the population/country where done. But country is mentioned for other studies in the same and ensuing paragraphs. Feels inconsistent.

We appreciate the suggestion and completed the text. Individuals included are Canadians and Americans; this information was included in line 101 as requested: (individuals Canadians and Americans were included in the analysis).

Methods

Generally very well explained.

We appreciate the comment.

I found the system of referring to base scenario then scenario 1 and scenario 2 somewhat confusing. Maybe rename so scenario 1 is base scenario or base scenario could be 0.

Thanks for the suggestion; however, we are concerned that Scenario 0 may be understood as a “business-as-usual” or baseline scenario, in which nothing has been done. We followed the time trend observed in the consumption of sugary beverages, as well as the time trend in the prevalence of obesity, both according to the adjustments used in the study, and therefore we chose to keep it as the baseline scenario, as it refers to what comes next. occurring in recent years. We apologize and ask that you consider keeping the base scenario name.

Line 117- “database for 2019”. Does that mean the version released in 2019 or data for the year 2019?

This means that data was collected in 2019 and published in the report in 2020. We apologize for not being transparent and have adjusted the text on lines 164-166: The VIGITEL (Surveillance System for Risk and Protective Factors for Chronic Diseases by Telephone Survey) database 2019 and published in the 2020 report was used to conduct the study.

Line 193: The authors state: The effect of reducing energy intake of sugary drinks from changes in FoPNL disregarded saturated fats and sugars to estimate body weight variation. Can the authors explain how this might have under or overestimated the overall effects of FOPNL on obesity prevalence?

In the VIGITEL survey, the volunteer is asked about consuming soft drinks or artificial juices, using soft drinks as the standard caloric value. In the mathematical model, the delta used refers to energy. Other sugary drinks may have fat in their composition, but we did not consider them in our analyses. We agree that this is a limitation of our study and included a sentence about it in lines 535-538: “Furthermore, another limitation refers to the use of soda as the only sugary beverage in this study. Other sugary beverages may contain fats in their composition, which were not considered in this analysis, which could affect the results obtained in the modeling.” We appreciate this observation. 

Major: In modelling studies, given the nature of simulations and predictions it is best practice to run sensitivity analyses around the assumptions made. This is a major omission from this paper. The authors chose Acton et al for estimating the effect of the FOPNL on SS consumption. The reason is sound but given that a meta-analysis exists, the findings from the meta-analysis and/or studies from the analysis should be used as part of a sensitivity analysis in the model to determine the extent of the difference had a different paper/estimate been chosen.

We agree with your observation and we included a sensitivity analysis after the first review, described in Appendix. There were changes in the modeled scenarios, and we chose to use the evaluation of the policy implemented in Chile through the results of the study of Taillie LS, Bercholz M, Popkin B, Reyes M, Colchero MA, Corvalán C. - Changes in food purchases after the Chilean policies on food labelling, marketing, and sales in schools: a before and after study. Lancet Planet Heal. 2021;5: 526–533. doi:10.1016/S2542-5196(21)00172.

We chose to use the Acton et al. (2019) study in a sensitivity analysis (scenario 3) and associated with reformulation (scenario 4). This meta-analysis was developed with three studies (Acton R, Jones A, Kirkpatrick S, Roberto C, Hammond D. Taxes and front-of-package labels improve the healthiness of beverage and snack purchases: a randomized experimental marketplace. Int J Behav Nutr Phys Act. 2019; 16:46. https://doi.org/10.1186/s12966-019-0799-0 PMID: 31113448; Acton R, Hammond D. The impact of price and nutrition labelling on sugary drink purchases: results from an experimental marketplace study. Appetite. 2018; 121:129–37. https://doi.org/10.1016/j.appet. 2017.11.089 PMID: 29146460; Grummon AH, Hall MG, Taillie LS, Brewer NT. How should sugar-sweetened beverage health warnings be designed? A randomized experiment. Prev Med. 2019; 121:158–66. https://doi.org/10.1016/j. ypmed.2019.02.010 PMID: 30772370). The study conducted by Acton et al. (2019) had the highest weight in the meta-analysis, greater than 70%, as shown in the figure below (figure in Response to reviewers, anexed).

This figure can be found in the supplementary material of the meta-analysis (https://doi.org/10.1371/journal.pmed.1003120). We used a subsample of Acton et al. (2019) study with the same age group as our study. Therefore, we chose to keep the Acton et al. (2019) study in our modeling scenarios and not use the meta-analysis result due to age group adjustment.

Results

Once again, generally well presented with a few queries:

We appreciate the comment.

Table 2 and the associated text line 298-304. The text states that the mean BMI of analyzed individuals was 26.5kg/m2, which I assumed was their BMI at the beginning of the five-year modeling exercise. In the Table, it states that the BMI in base scenario is 26.5. This confused me because I thought the mean BMI of 26.5 referred to the mean BMI before any scenarios were applied. Thus I am expecting that there should be a figure that estimates the change in BMI under the base scenario where change in consumption is reduced by -20.

We thank you for your observation and added 2019 to Table 2 in the first and second lines. The remaining rows in Table 2 are the estimates after five years of FoPNL implementation. 

We used the base scenario to calculate the others: we simulated scenarios 1 and 2 and made the difference with the base scenario. After that, we simulated the prevalence of obesity (time trend) and used the previous result to estimate the reduction in cases of obesity. The base scenario is part of the calculation.

In other worse, the base scenario represents a reduction in consumption of -20 after 5 years (Table 1) thus there should also be some reduction in BMI from this just as there is for scenario 1 where the expected change after 5 years is. Although the consumption of sugary beverages in Brazil is decreasing, the prevalence of obesity is increasing, and these factors were considered in all the scenarios. For this reason, we thought it made no sense to present the result of the base scenario in isolation regarding this reduction of 20kcal/per person/day.

Line 306- Figures 1: C1, C2 and C3 do not show a reduction in obesity prevalence and so appears to contradict the text. I believe the authors are referring to the reduction in scenario 1 relative to the base scenario. That is not stated in text and the particular graphs being referred to is not clear.

The modeled impact of FoPNL implementation refers to a slowdown in the increase in the prevalence of obesity. It would not be capable of reversing the problem in the country alone. This study was limited to simulating the impact of a single category of beverages on the obesity outcome in the country. We complete this excerpt with the sentence below (lines 384-386): “However, the reduction of simulated cases is unable to incline the curve of the increase in the prevalence of obesity, but it has the potential to slow down the growth.”

Line 308 refers to Figure 1 but there are six graphs attached to Figure 1. Is the 0.25 to be read from Graph A, B-1 B-2 or the series of C graphs?

We agree and appreciate the suggestion. In lines 378 and 380, we included (Fig 1, B-1) and (Fig 1, D-1), respectively.

Line 308- Can you clarify that ‘pp’ refers to percentage points? It does not appear in the graphs only in text.

Yes, it refers to percentage points. We changed all the ‘pp’ for percentage points throughout the text. 

The reference in line 310 to numbers of cases of obesity avoided after FOPNL is more intuitive and informative than the percentage points more commonly referred to by authors. I recommend more use of this. Perhaps use this in the abstract to make it easier for policy makers to understand and provide more concrete thinking in terms of the impact of the FOPNL.

We included this information really to give a clear idea of the number of people who would benefit from the implementation of FoPNL. We appreciate the comment.

Discussion

This was generally very long with information that reached outside the scope of the paper insufficient focus on discussing the limitation such as the potential limitation arising from line 193.

We appreciate this observation and we included the following sentence to lines 535-538: “Furthermore, another limitation refers to the use of soda as the only sugary beverage in this study. Other sugary beverage may contain fats in their composition, which were not considered in this analysis, which could affect the results obtained in the modeling.”.

For example, the two pages spent explaining Chile’s journey through all obesity prevention policies could be summarized into one paragraph which focuses on the impact FOPNL and the multiple policies has had on obesity/overweight prevalence since that is the focus of this paper.

We thank you for this comment and agree with it. This was included in the re-discussion of the entire article in the first review.

7. PLOS authors have the option to publish the peer review history of their article (what does this mean?). If published, this will include your full peer review and any attached files.

Do you want your identity to be public for this peer review? For information about this choice, including consent withdrawal, please see our Privacy Policy.

Reviewer #2: No

Reviewer #3: No

---

## [Editor Report · Decision Letter 2]

18 Jul 2023

Impact of implementation of front-of-package nutrition labeling on sugary beverage consumption and consequently on the prevalence of excess body weight and obesity and related direct costs in Brazil: An estimate through a modeling study

PONE-D-22-26938R2

Dear Dr. Lucilene Rezende Anastácio

We’re pleased to inform you that your manuscript has been judged scientifically suitable for publication and will be formally accepted for publication once it meets all outstanding technical requirements.

Kind regards,

Anselm J. M. Hennis

Academic Editor

PLOS ONE

---

## [Editor Report · Acceptance letter]

2 Aug 2023

PONE-D-22-26938R2 

Impact of implementation of front-of-package nutrition labeling on sugary beverage consumption and consequently on the prevalence of excess body weight and obesity and related direct costs in Brazil: An estimate through a modeling study 

Dear Dr. Anastácio:

I'm pleased to inform you that your manuscript has been deemed suitable for publication in PLOS ONE. Congratulations! Your manuscript is now with our production department. 

Kind regards, 

on behalf of

Dr. Anselm J. M. Hennis 

Academic Editor

PLOS ONE